Methods

# An integrative machine learning approach for prediction of toxicity-related drug safety

Artem Lysenko[1] , Alok Sharma[1,2], Keith A Boroevich[1] , Tatsuhiko Tsunoda[1,3,4]

**Recent trends in drug development have been marked by diminishing returns caused by the escalating costs and falling rates of new drug approval. Unacceptable drug toxicity is a substantial cause of drug failure during clinical trials and the leading cause of drug withdraws after release to the market. Computational methods capable of predicting these failures can reduce the waste of resources and time devoted to the investigation of compounds that ultimately fail. We propose an original machine learning method that leverages identity of drug targets and off-targets, functional impact score computed from Gene Ontology annotations, and biological network data to predict drug toxicity. We demonstrate that our method (TargeTox) can distinguish potentially idiosyncratically toxic drugs from safe drugs and is also suitable for speculative evaluation of different target sets to support the design of optimal low-toxicity combinations.**

## Introduction

The last decade has seen an escalation of drug development costs, and at the same time, the rate at which new successful drugs are released has actually decreased (1). One striking example of this trend was put forward by (2), who observed that during the period of 2004–2014, both the funding and number of drug candidates trialed in the United States increased substantially, but the number of new drugs approved declined by more than 25% compared with the previous decade. Unacceptably high toxicity is a major contributing cause of drug failure and accounts for about one-fifth of clinical trial failures (3) and two-thirds of worldwide post-launch withdrawals (4). One strategy to reduce these costs and improve the efficiency of the drug development is to augment laboratory and clinical testing with computational analysis (5), and the development of accurate methods to predict toxicity is pivotal to this goal (6).

Earlier methods for computational pre-screening focused primarily on chemical features of potential compounds. The first approaches were frequently based on rule sets (7) with scores awarded to compounds for not failing particular criteria of "drug-likeness". From the pharmacokinetic perspective, it was proposed to characterize compounds according to absorption, distribution, metabolism, and excretion criteria (ADME) (8). Further developments have led to refinements of simple rule-based methods into more granular qualitative measures, such as quantitative estimate for drug-likeness (QED) (9), which uses a desirability function to compute an optimal score across multiple chemistry-based criteria. Importantly, most of these efforts were not specifically intended only to identify likely toxicity, but to also optimize over a range of relevant properties that can impact efficacy, bioavailability, and pharmacokinetics.

A recent evaluation of current methods was performed by (10). Their work has shown that chemistry-based scoring and rule-based systems have only very modest power to predict clinical trial failures. These methods could not accurately predict clinical trial failure due to drug toxicity if taken in isolation and not combined with additional features. One possible explanation is that these schemes, like Lipinski's Rule of 5 (11), are now routinely used to screen drugs (12), and compounds at clinical trial stage are likely to have already passed such screening. Another part of the explanation could be that these rules do not strongly apply to a very large subset (estimated at 50–80% of all drugs) of "metabolite-like" compounds that can mimic naturally occurring metabolites and behave in a similar way (13). Lastly, toxicity-related responses are mostly mediated by drug–protein interactions (14), which may not necessarily have a clear correspondence to molecular structure features.

Given the complex nature of the drug toxicity prediction problem, the chemistry-led approaches outlined above are just one of the many possible ways to consider it, and other studies have explored a wide variety of alternative strategies. Several proposed methods have used various semantic similarity (15) and correlation measures, such as known side-effect profiles (16, 17) to predict specific side-effect labels. An alternative perspective was developed by (18), which used predictive binding to a small number of already known toxicity-related proteins as an indicator of risk. Yet another set of strategies rely on leveraging gene expression (19, 20) and metabolomics profile similarities (21). Although all of these

[1]Laboratory for Medical Science Mathematics, Rikagaku Kenkyūjyo Center for Integrative Medical Sciences, Tsurumi, Japan    [2]School of Engineering and Physics, University of the South Pacific, Suva, Fiji    [3]Department of Medical Science Mathematics, Medical Research Institute, Tokyo Medical and Dental University, Tokyo, Japan    [4]Core Research for Evolutionary Science and Technology Program, Japan Science and Technology Agency, Tokyo, Japan

Correspondence: artem.lysenko@riken.jp; tsunoda.mesm@mri.tmd.ac.jp

works have undoubtedly greatly contributed to our understanding of the patterns and mechanisms of toxic side effects, not all of these approaches can be used during drug the development process because large amounts of in vivo, human-specific data can only be safely collected once the risks of the candidate drug are sufficiently understood.

Another complication arises when attempting to integrate or fairly compare approaches because, often, the scope or prediction goals of different methods are not readily comparable. Frequently specialized methods are developed for particular classes of compounds (22) or specific, carefully defined scenarios (23, 24). Although most methods measure their success in terms of their ability to predict all possible types of side effects (i.e., from relatively benign to highly dangerous), others (10, 25) consider drug toxicity in terms of drug trail failures or withdrawals from the market—a criteria most similar to the one used in this study. Clearly "drug rejection" criteria are not directly comparable with the "side effect prediction" criteria, as in the former case, most dangerous side effects are prioritized and "unsafe" category assignment is indirectly affected by factors such as the overall severity and frequency of toxic responses and ability to effectively manage those risks. Although both (10) and (25) used comparable criteria, the latter made extensive use of drug annotation (phenotypic indications and all known adverse reactions). Such data can only be collected once the drug is in use for some time and is not available for new compounds, such as novel candidate drugs from the ClinicalTrials.gov database used in this study.

Drug toxicity is commonly classified into two subtypes: Type A or intrinsic toxicity, which is dose dependent and related to the primary pharmacological target of the drug, and Type B or idiosyncratic toxicity (IT), which is unpredictable, occurs at frequencies of less than 1 in 5,000 cases (26), is not dose dependent, and is associated with off-target effects (27). Although decisions to withdraw a drug from the market can be made for a variety of reasons, unacceptable IT is believed to be the main reason (28, 29, 30, 31). Given that IT is very rare, it can be unnoticeable in smaller test populations used for clinical trials and is often not detectable in animal models (27). Our analysis indicated that current leading methods developed for clinical trial success prediction and drug-likeness do not perform as well in the case of drugs withdrawn from the market (Fig S1). This may indicate that a different perspective or drug properties are needed to specifically capture those effects. However, we would like to emphasize that data used to develop these tools and their goals were not optimized for prediction of drug withdrawals from the market, and the result reported here is by no means representative of the performance of these tools in their intended contexts.

Motivated by the importance of drug–protein interactions in drug toxicity mechanisms (14) and the increasing prominence of target-based drug development, in this study, we explore the feasibility of developing a computational target-driven drug toxicity prediction method (TargeTox). The method uses information about all proteins that can bind a drug (both intended pharmacological targets and off-targets) in combination with machine learning to identify potentially toxic compounds. Importantly, drugs can have both type A and type B toxicity risks at the same time, and therefore, a combination of these factors can lead to the conclusion that particular

drug is unsafe. At present, no relevant databases provide structured and comprehensive information about type A and type B toxicity risks; however, it is generally believed that type A toxicity is predominantly discovered during clinical trials and type B during the monitoring and reporting stage after release to the market (27). For these reasons, when designing a training dataset, we aimed to include examples for both cases, although in our downstream analysis, we place particular emphasis on confirming performance for IT cases. Although we aim to predict toxicity risk of both types, current implementation is not designed to directly identify which type is prevalent for specific cases.

One particular challenge in the incorporation of drug-binding protein data is the sparse nature of the dataset, where each drug will only bind a relatively small set of all possible proteins and this number will greatly differ between drugs. At the same time, given that the set of confirmed toxic drugs satisfying our criteria is small, large numbers of bound proteins and most interactions will occur only once in the entire dataset. To address this, we propose to leverage a guilt-by-association principle in combination with the biological network context of these proteins. Because it was previously reported that target proximity in the network corresponds to the similarity of drug side effects (32), we hypothesized that severe toxicity-related responses could be localized to particular regions of the biological network. Here, the network is represented by a distance matrix of all constituent proteins. Our analysis has shown that both simpler and more sophisticated network distance measures can be used with this approach; although based on our evaluations, diffusion state distance (DSD) (33) was chosen as the marginally better performing metric. The position of each drug-specific set of bound proteins is approximately encoded by distances to a small number of reference proteins. This interpretation of the data allows all observations to be meaningfully used, including cases where single instances of drug-binding proteins are found in training or evaluation datasets. Although concepts of biological network diffusion have been explored in other contexts, for example, as in (34), the distinguishing and novel feature of our approach is the direct use of a machine learning classifier to "map out" areas of the network during the training process, which means other covariates can also be taken into account in conjunction with network-based location data. At the same time, this method can also reduce dimensionality and convert data from sparse to dense representation.

# Results

## Drug-binding proteins tend to be non-uniformly distributed in the network

Information for all drug-binding proteins in our reference set was acquired from the DrugBank and ChEMBL databases. Although there was a substantial number of drugs with a single target (Fig 1), most drugs interacted with more than one protein. The number of bound proteins was also smaller than the number of drugs, and about 47% of all these proteins were found in both toxic and safe subsets. To explore the overall distribution of drug-binding proteins in the context of the human interactome they were combined

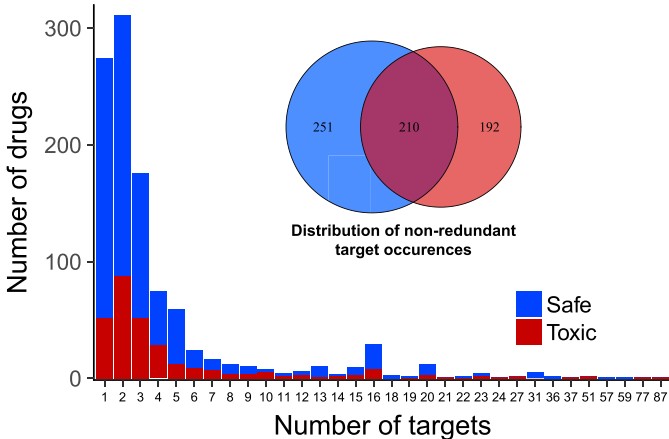

**Figure 1. Distribution of drug-binding proteins for all drugs in "toxic" and "safe" categories.**
Counts of drugs with a particular number of bound proteins (main/off-target) (*n* = 893), above—Venn diagram shows how many distinct proteins were bound by at least one drug from each subset.

with a protein association network from the STRING database, which was transformed into a DSD matrix to do this analysis. Overall DSD distribution for all proteins in the main connected component of the network was largely consistent with what was previously reported by (33) for the yeast protein–protein interaction network (Fig 2A). The generated distribution had a relatively smooth central part with a long right tail. To visually explore possible location patterns of bound proteins, we mapped the complete DSD distance matrix into two dimensions (Fig 2B) using the t-distributed stochastic neighbor embedding (t-SNE) algorithm (35). Although a minority of bound proteins appeared to be dispersed throughout the network, most tended to be co-located in a few distinctive groups. On average, bound proteins of the same drug tended to be significantly closer together than random samples of the same size, although there was no difference between average distances of toxic and safe drug–interacting protein sets (Fig 3A) and the same pattern was observed when only a subset of drugs withdrawn from the market was considered (Fig 3B). The overall proximity of the proteins binding the same drug suggested a possibility that these sets may be represented more compactly by network locations to

reduce the sparseness and dimensionality of the data while minimizing the loss of useful information.

## Computational model for prediction of dangerous drug toxicity

To facilitate accurate identification of potentially toxic drug candidates, we have developed a Biological Network Target-based Drug Toxicity Risk Prediction method (TargeTox). The method aims to leverage the guilt-by-association principle, according to which entities close to each other in biological networks tend to share functional roles. The distance between nodes in biological networks can be quantified using a variety of different methods, and we have evaluated several strategies ranging from very simple approaches, such as the shortest path method, to novel and advanced approaches, such as the Mashup (36) method that integrates diffusion-based distances across multiple 'omics networks. By interpreting a network as a set of pairwise distances, biological functions and phenotypes can be associated with areas of the network rather than just individual nodes and their location can be efficiently summarized with respect to a few reference points. Once the network location data have been put into this form and combined with relevant covariates, a machine learning classifier was trained on the combined dataset. In principle, this strategy can be used in combination with any modern classifier that has some form of regularization capabilities and can handle non-linear relationships, for example, certain SVM variants or deep neural networks. However, in this case, the gradient-boosted classifier tree ensemble model (GBM) was chosen for the following two reasons: First, given the small number of positive (toxic) drugs in our training dataset, the comparatively less hyper-parameter tuning needed by the GBM was particularly helpful for mitigating the over-fitting risk. Second, GBM can handle the presence of missing values in our data without the need for prior imputation, thereby greatly simplifying both development and any possible future applications of our method.

To control for the risk of over-fitting our model, the available data were split into a training set (80% of all drugs, Fig 4) and a hold-out validation set (20%). The performance of different design strategies and hyper-parameter configurations was evaluated on the training set using five-fold cross-validation, then a model was

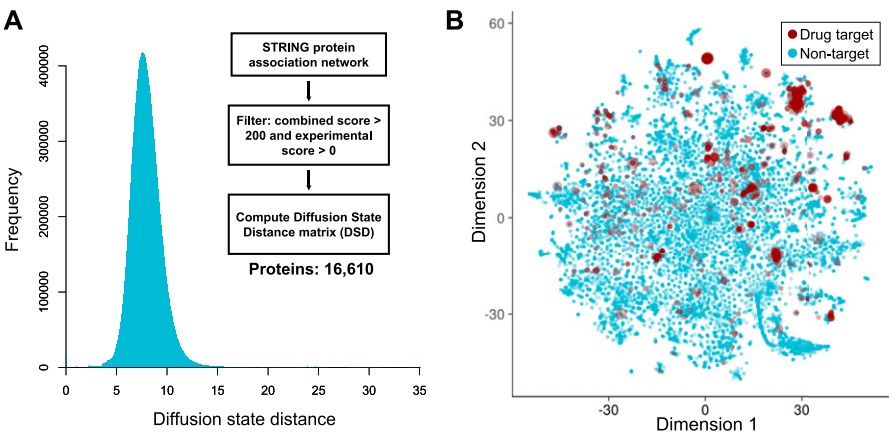

**Figure 2. DSDs of the protein–protein interaction network.**
**(A)** Overall distribution of all pairwise DSDs for the main connected component of the network; flow chart shows an overview of the analysis used to convert the STRING protein–protein interaction network into a DSD matrix. **(B)** Relative positions of all proteins (*n* = 16,610) in the DSD space. All pairwise distances were projected into two dimensions using the t-distributed stochastic neighbor embedding (t-SNE) algorithm. Red circles show all drug-binding targets and the size of the circle is proportional to the number of different drugs targeting that protein.

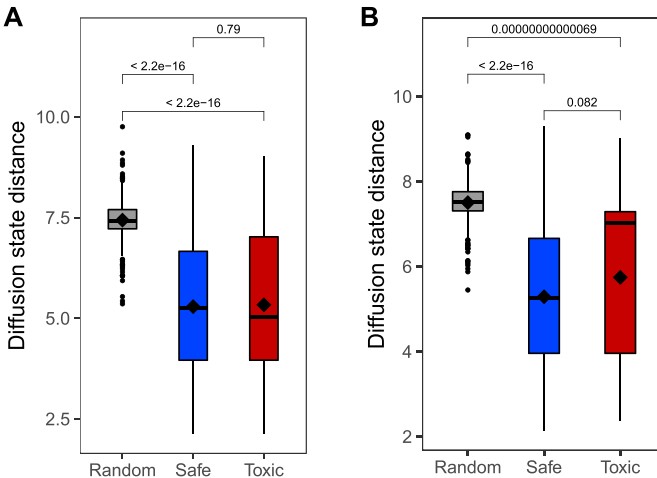

**Figure 3. Comparison of distances between proteins binding the same drug and a random sample.**
**(A)** All "toxic" and "safe" drugs versus an analogous random sample. **(B)** All "safe" drugs versus a subset of the "toxic" set that were withdrawn from the market and an analogous random sample. In both cases the significance was computed using Wilcoxon signed-ranks test.

trained on the complete training set and evaluated on the remaining data. We evaluated the following strategies for measuring network distances: shortest path, discretized shortest path (1 if less than length 3, 0 otherwise), DSD (33), and mashup-based method (36). Our evaluation results showed that, generally, all of the tested measures can to some extent be used in combination with our method. In addition, we have evaluated two other ways of summarizing drug-binding protein information: (1) using a medoid protein for a set of all proteins binding a particular drug and (2) using distance to all other proteins in the network rather than choosing a few reference proteins. The first strategy had achieved 69.69% receiver–operator curve (ROC) AUC (Fig S2A). The second strategy was the second-best performing of all the tested configurations (ROC AUC of 72.79%, Fig S2B), however at a great cost of time needed to train the model. For the latter strategy, we also observed that in actuality only some points were used in the trained model as GBM algorithm performs feature selection during training. Compared to those, the simple shortest path version had only slightly lower performance (Fig S2C) and the discretized shortest path version had the lowest overall ROC AUC of 68.4% (Fig S2D). The performance of the method variant using mashup-based distances was better than that of the shortest path version but still lower than that of the DSD-based approach (Fig 5A). The best strategy was to use a small number of reference points with a DSD metric, and for each of them take a distance to the closest protein bound by a given drug. This method achieved ROC AUC of 73.4% on the training subset (Fig 5B) and, at optimal trade-off point, had a sensitivity of 74.7 and a specificity of 65.8. On hold-out test set, ROC AUC for this optimal version was 71.30% (Fig 5C). Feature importance analysis performed on the final version of the model (Fig 5D) indicated that, in aggregate, network-based features were the most important category accounting for half of all importance, whereas functional impact (FI) was the most important single feature. No features were discarded as a result of feature selection performed by the algorithm during training.

## Evaluation of ability to predict IT

To investigate the potential of the method to detect idiosyncratically toxic drugs, we have identified two relevant subsets. The first had 38 drugs from our dataset that have been specifically identified as idiosyncratically toxic in the literature. The second subset had nine drugs associated with HLA-mediated toxicity (37). HLA-mediated toxicity is one of the prominent and relatively well-studied examples of IT. Therefore, we reasoned that these drugs could be used to explore the ability of our approach to identify potential common toxicity mechanisms for a group of drugs.

To explore the performance for the more general set of 38 idiosyncratic toxic drugs, we performed a leave-one-out cross-validation and compared the scores of the 38 drugs with those in the safe subset. The scores were consistently and significantly higher (i.e., predicted to be more toxic) for this subset (Fig 5E). Then, we performed the same comparison for the scores from PrOCTOR and weighted QED methods, but no significant differences were detected for either method (Fig S3). To explore whether our chosen features could capture patterns specific to an idiosyncratic subset, a more detailed feature attribution analysis was performed using the SHAP (shapley additive explanation) value methodology for gradient-boosted tree ensembles (38 *Preprint*). After computing feature-specific SHAP values for each drug, we compared an idiosyncratically toxic subset with drugs where toxicity was identified during clinical trials. For the latter subset, we verified that IT was not reported as the main cause of clinical trial termination in the corresponding entry of the ClinicalTrails.gov database. In addition,

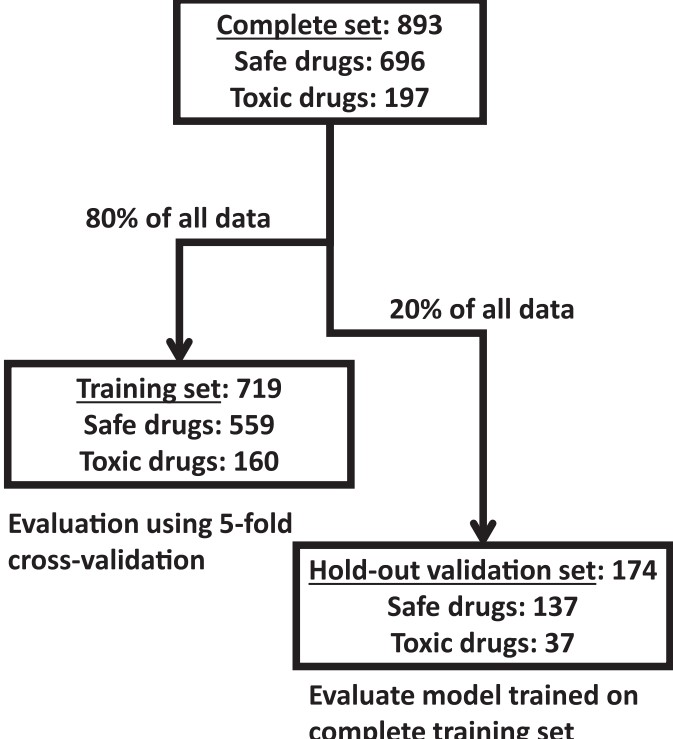

**Figure 4. Overall composition of the selected drugs dataset and its partitioning for model development.**

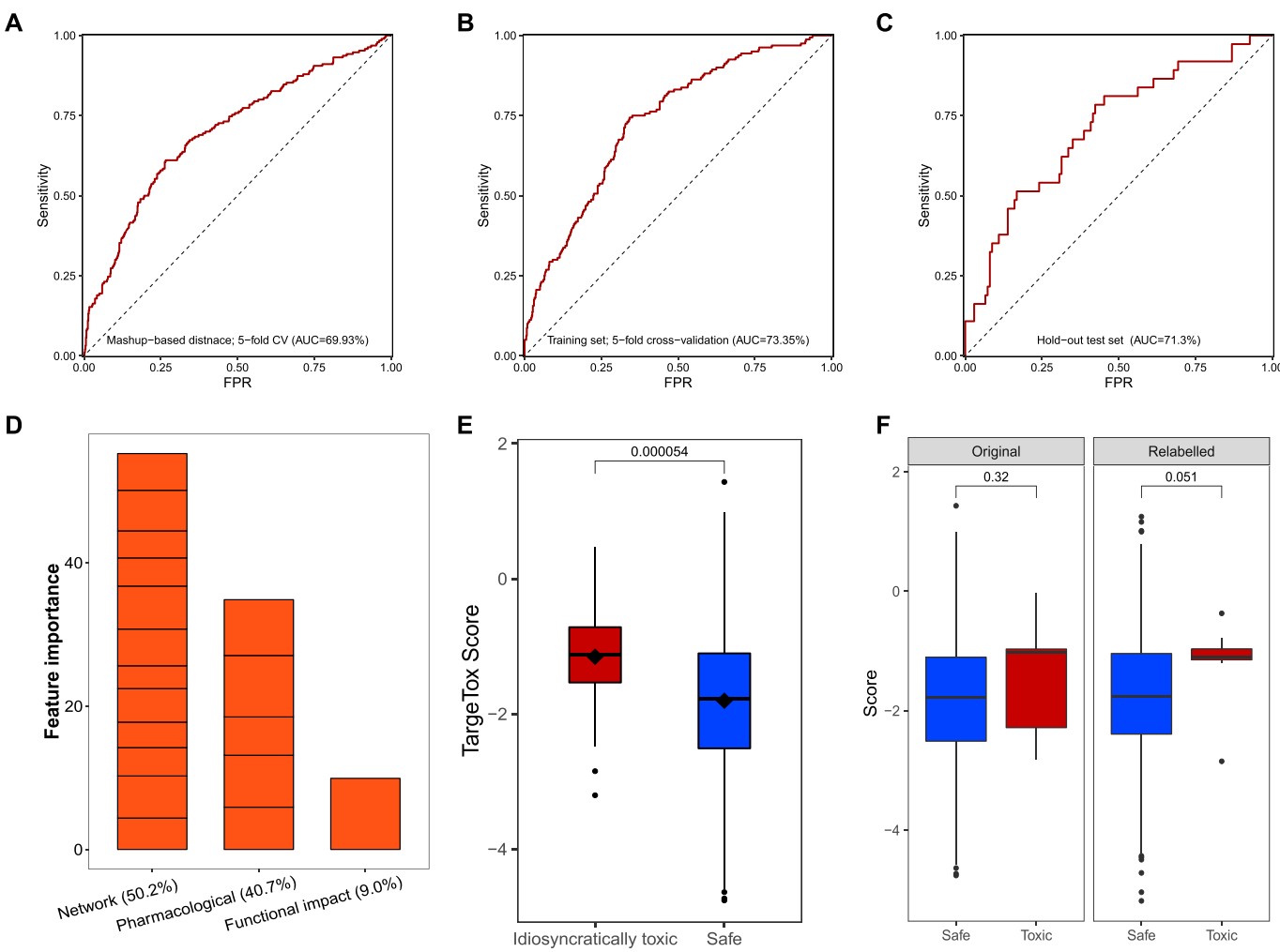

**Figure 5. Performance benchmarks and feature importance analysis.**
Receiver-operator characteristic (ROC) curves for different model variants. **(A)** Mashup-based distance version evaluated on the training set using five-fold cross-validation (CV) (*n* = 719). **(B)** DSD version evaluated on the training set using five-fold CV (*n* = 719). **(C)** DSD version validated on the hold-out set (*n* = 174). **(D)** Contribution of different features to the model measured in relative feature importance. **(E)** Comparison of scores returned by the model for the idiosyncratically toxic (*n* = 38) and safe subsets (*n* = 696) computed using the final model and leave-one-out cross-validation. **(F)** Comparison of scores for toxic drugs linked to HLA-mediated toxicity (*n* = 9) computed using leave-one-out cross-validation; "original" sub-plot shows scores if curation-based toxicity annotation was used, and "relabelled" sub-plot shows scores when all relevant drugs are relabeled as toxic. Significance was computed using Wilcoxon signed-ranks test.

to the best of our ability, we checked for other factors that could bias these results, including over-representation of particular drug classes or indications. Each pair of SHAP value distributions was compared using Wilcoxon signed-ranks test that identified seven significant differences at 5%, of which one was also significant at the 1% level (Table 1). These results suggest that a substantial number of features identified as particularly important for type B versus type A toxicity are distinctive and these differences were captured by our method design.

To explore which features were used to correctly classify drugs in an idiosyncratically toxic subset, we compared the relative positive SHAP score allocations toward all model features (Fig S4). The main difference was in the greater weight placed on all of the pharmacologic features (two plasma protein binding and three route of administration features). FI score also had about 3% higher relative SHAP importance, whereas importance of the network-based

feature category decreased by about 11%. There were also considerable re-allocations of importance within the network category itself, indicating that different parts of the network were important for correct classification of these two groups of drugs.

For another evaluation we identified nine drugs known to be idiosyncratically toxic via a HLA-mediated mechanism. Of all the drugs in this category, only three were already categorized as toxic according to our chosen criteria (clinical trial failure or market withdrawal for reasons of toxicity). One possible explanation could be that given that this particular mechanism is well-researched, effective strategies exist (e.g., known risk alleles, populations and treatment regimens) to manage these risk allowing most drugs to be used relatively safely. Similarly, leave-one-out cross-validation (Fig 5F, left) using original safe/toxic assignment did not indicate that these drugs were significantly more toxic that the main "safe" category. Next, to further explore the potential of our method to

**Table 1. Comparative shapley additive explanation (SHAP) analysis of model feature importance.**

| Feature | Average SHAP value | | Wilcoxon test *P*-value |
|---|---|---|---|
| | Idiosyncratic toxicity | Clinical trial toxicity | |
| Network-based features | | | |
| 1 | 0.008 | 0.068 | 0.106 |
| 2 | 0.043 | −0.006 | 0.065 |
| 3 | 0.003 | −0.006 | 0.438 |
| 4 | 0.021 | −0.012 | 0.025* |
| 5 | 0.026 | 0.071 | 0.798 |
| 6 | −0.004 | −0.002 | 0.450 |
| 7 | 0.001 | 0.166 | 0.003** |
| 8 | −0.014 | 0.255 | 0.017* |
| 9 | 0.005 | −0.005 | 0.044* |
| 10 | 0.003 | 0.043 | 0.062 |
| 11 | 0.058 | 0.009 | 0.019* |
| 12 | 0.047 | 0.052 | 0.659 |
| Functional diversity | 0.104 | 0.181 | 0.601 |
| Administration route | | | |
| Oral | 0.047 | 0.036 | 0.554 |
| Parenteral | 0.043 | 0.014 | 0.019* |
| Topical | 0.033 | 0.026 | 0.674 |
| Protein binding | | | |
| Lower bound | 0.025 | 0.146 | 0.01* |
| Upper bound | 0.014 | 0.097 | 0.171 |

*$P < 0.05$; **$P < 0.01$.

detect common toxicity mechanism of this group of drugs, we conducted an additional leave-one-out validation where all nine drugs were relabeled as "toxic". This change caused an increase in the predicted toxicity score for most members of the set; however, overall, this difference was not found to be significant at the 5% level (Fig 5F, right).

### Independent validation using side-effect annotation

A secondary validation was performed using side-effect annotation from the OFFSIDES database (39), from which we selected drugs not present in either the training or hold-out subsets. After pre-processing, the validation dataset contained 339 drugs. Given the wide scope and diversity of possible side effects, many of which are not considered severe enough to preclude the use of a drug, we have selected 14 toxicity-related categories commonly associated with failed drugs, including cardiotoxicity, hepatotoxicity, and toxic shock. Predicted scores of drugs in these subsets were compared with a set of 120 compounds that did not have any of these annotations (Table 2). The average score of these major toxicity-associated categories was higher than the average of the un-annotated set, and the difference was significant in all individual

**Table 2. Average scores of the OFFSIDES toxicity categories compared with 120 drugs without such annotations.**

| OFFSIDES side effect | Counts | Mean TargeTox score | *P*-value |
|---|---|---|---|
| Cardiotoxicity | 26 | 1.12 | $2.257 \times 10^{-6}$ |
| Skin toxicity | 30 | 0.95 | $1.180 \times 10^{-5}$ |
| Pulmonary toxicity | 38 | 0.69 | $1.425 \times 10^{-4}$ |
| Gastrointestinal toxicity | 34 | 0.73 | $4.766 \times 10^{-4}$ |
| Toxic encephalopathy | 46 | 0.40 | $1.249 \times 10^{-3}$ |
| Haematotoxicity | 39 | 0.56 | $1.112 \times 10^{-3}$ |
| Hepatotoxicity | 66 | 0.15 | $5.588 \times 10^{-3}$ |
| Ocular toxicity | 12 | 1.04 | $1.891 \times 10^{-3}$ |
| Bone marrow toxicity | 23 | 0.44 | $9.885 \times 10^{-3}$ |
| Toxic shock | 10 | 0.75 | 0.011 |
| Drug toxicity | 120 | 0.11 | 0.011 |
| Ototoxicity | 15 | 0.64 | 0.023 |
| Nephropathy toxic | 47 | −0.05 | 0.184 |
| Mitochondrial toxicity | 15 | −0.09 | 0.266 |

cases except for nephropathy toxic and mitochondrial toxicity categories. Likewise, the overall difference of the pooled set had a significantly higher average score (Fig 6). These results reaffirm the particular relevance of the proposed method for identification of high-risk drugs of these types.

### Model interpretation

Although gradient-boosted tree ensemble methods are very powerful and flexible, the complexity of generated models makes them challenging to interpret directly. An additional complication arising from the chosen design of network-based features is that by itself the individual importance of a reference protein feature may not directly identify bound proteins associated with toxicity risk, but rather the approximate location of the relevant proteins in the network. In some cases, this location can only be defined by a higher order interaction of several such features. Nevertheless, this information is captured by the model and can be recovered to profile the potential toxicity risk of different proteins.

To extract this overall "toxicity risk map" from the model, we created a simulated dataset of single-target drugs for each of the proteins in the "druggable genome" list from the work of (40). Most of this set (4,019 proteins) could be mapped to the main connected component of the STRING protein-association network. Notably, the highest score achieved by a simulated single-target drug was 45% lower than the top score in a real dataset, indicating that the highest predicted toxicity risks are because of the combined effect of multiple causal proteins. Despite this important difference, these results could still be useful for interpreting the behavior of the technically "black-box" model and extraction of informative insights. Distributions of the scores assigned to these proteins by TargeTox were visualized to check for the presence of the coherent structure. Again, this was performed with the aid of t-SNE to project the positions relative to the 12 reference points into two

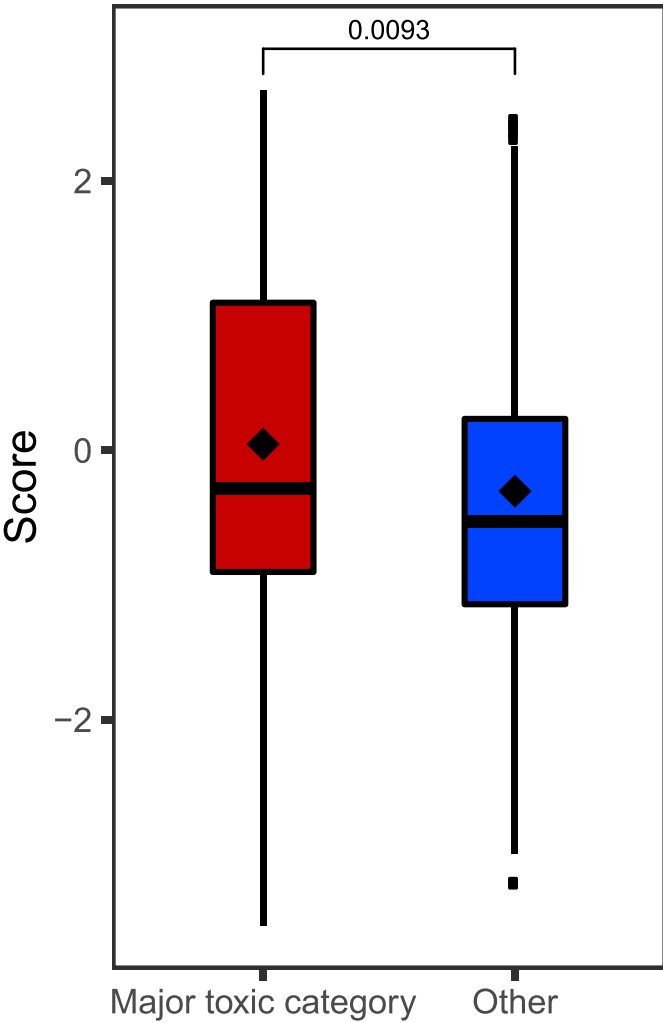

**Figure 6. Comparison of scores for annotation-based toxicity categories.** major toxic category contains all drugs with toxicity-related annotations from the OFFSIDES database (*n* = 257), and the safe category contains all drugs without any such annotations (*n* = 120). All drugs that were already present in a set used to train the model were excluded. Significance was computed using Wilcoxon signed-ranks test.

dimensions. These results (Fig 7A) showed that bound proteins predicted to have a higher risk are concentrated in several hot spots. The top 10% of these predictions were separately clustered to identify possible high toxicity risk subgroups. Clustering suggested the presence of eight subgroups (Fig 7B), four of which (1–3 and 6) corresponded to compact and distinctive neighborhoods suggested by the t-SNE algorithm and four others were distributed over wider areas.

Common functional roles of these protein groups were identified using functional enrichment analysis (Fisher's exact test with false discovery rate correction) with respect to the biological process (BP) aspect of the Gene Ontology. Overall, the most common recurring processes included signaling and protein phosphorylation, with multiple significant hits across all clusters, with the highest fraction in cluster 1 (94.12% of all proteins, $P$ = 0.002). The highest predicted toxicity risk scores were particularly concentrated in

clusters 1 through 3, which were also placed close together by the t-SNE algorithm, suggesting similar protein–protein interaction context. Some notable potentially relevant functions included immune-related processes (clusters 1, 2, 5, and 6, or multiple). Disruption of immune system processes frequently underlies toxic side effects (41). Clusters 1 and 3 were enriched for peptidyl-tyrosine phosphorylation (86.3% of all members, $P = 4.02 \times 10^{-30}$ in cluster 1 and 70% in cluster 3, $P$ = 0.002) and, in cluster 1, also positive regulation of JAK-STAT cascade (13.73%, $P = 2.74 \times 10^{-4}$). Tyrosine kinases are prominently linked to idiosyncratic toxic side effects, including cardiotoxicity (42), whereas the *JAK-STAT* pathway is important for different aspects of neurologic toxicity (43). Cluster 5 was significantly enriched for response to toxins (15.38%, $P$ = 0.03). Cluster 4 had a high number of G-protein–coupled receptor signaling pathway members (60%, $P$ = 0.01). Apoptotic process, believed to play an important role in drug-induced hepatotoxicity (44) and cardiotoxicity (45), was the largest enriched category in cluster 7 (63.62%, $P$ = 0.01). No significant GO term enrichment for any functions was identified in cluster 8. Full results of this analysis in the form of gene annotations, their cluster assignments, and GO BP enrichment are provided in the supplementary material (Tables S1 and S2).

In terms of individual protein ranking of the "druggable genome" set, the highest toxicity-scoring predictions were concentrated in clusters 1–3. Top predictions featured several proteins identified as promising anti-cancer drug targets. The highest ranked protein with a score of 1.77 was FGFR2, a tyrosine kinase and an important oncogene. In particular, this protein binds the AZD4547 candidate drug, clinical trials of which have reported a number of serious toxicity incidents (46). The third highest scoring protein TLR4 is suggested to play an important role in chemotherapy-induced gut toxicity (47) and nephrotoxicity (48). Among other proteins in the top 10 were AKT1, KIT, JAK2, and LYN. AKT1 is a serine/threonine kinase, inhibition of which was found to be linked to liver injury and development of hepatocellular carcinoma in animal models, with possible implications for human clinical trials currently in progress (49). The proteins KIT, JAK2, and LYN are all members of the tyrosine kinase family that have many promising drug targets while also being associated with serious toxicities, both on-target (50) and unexpected (51), and, more specifically, idiosyncratic hepatotoxicity (52). One notable example somewhat further down the list was PTGS2 (COX2), which was rated in the top 2% for likely toxicity and is the key protein target, leading to high-profile withdrawal of Vioxx drug because of the doubling of heart attack risk (53).

## Discussion

One principle obstacle to the development of computational predictive approaches is the sheer complexity of the interplay between factors that determine whether a drug is considered to have an unacceptable level of toxicity (Fig 8). The fundamental trade-off at the core of this decision is striking the right balance between efficacy and safety (54), both of which are evaluated with respect to the severity of the disease to be treated. Each of these factors may be subject to considerable variation. Only a small number of

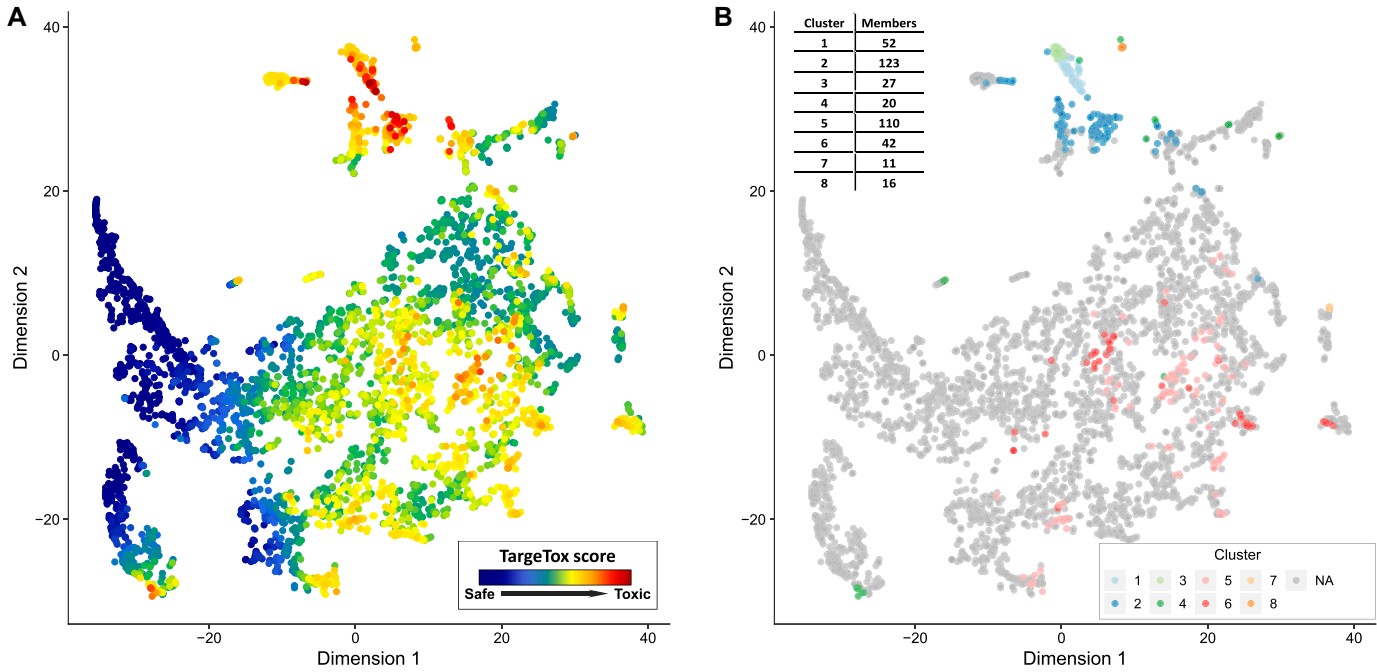

**Figure 7.  Druggable proteome annotation with the TargeTox method.**
Two panels show the druggable proteome (*n* = 4,019) with layout based on DSDs computed from the STRING protein association network and mapped to two dimensions using the t-SNE method. **(A)** Protein nodes colored according to TargeTox score (red = highest risk, blue = lowest risk). **(B)** Locations of distinctive subgroups with highest risk (top 10% of all druggable proteome by TargeTox score) with groups derived by clustering their DSD vectors.

people might experience a toxic side effect and it may have different degrees of severity and, in the case of idiosyncratic responses, the exact underlying causes can be particularly complex (55). Furthermore, extrinsic factors such as cost, availability of alternative treatments, and ability to predict or manage risks also inform the final decision (56). Although ideally it is very desirable to directly incorporate the effects of these factors into a predictive toxicity model, at present such data are still not systematically collected at the necessary level of detail. Not being able to

accurately model these effects is an important factor limiting the accuracy of computational drug screening approaches, but structured data collection efforts by initiatives such as ClinicalTrials.gov are likely to address this data availability problem in the near future.

Another limitation is the actual number of failed drug observations that are currently available in the public domain. A very large number of features may need to be included in the model to adequately capture the underlying complexity, which in turn would

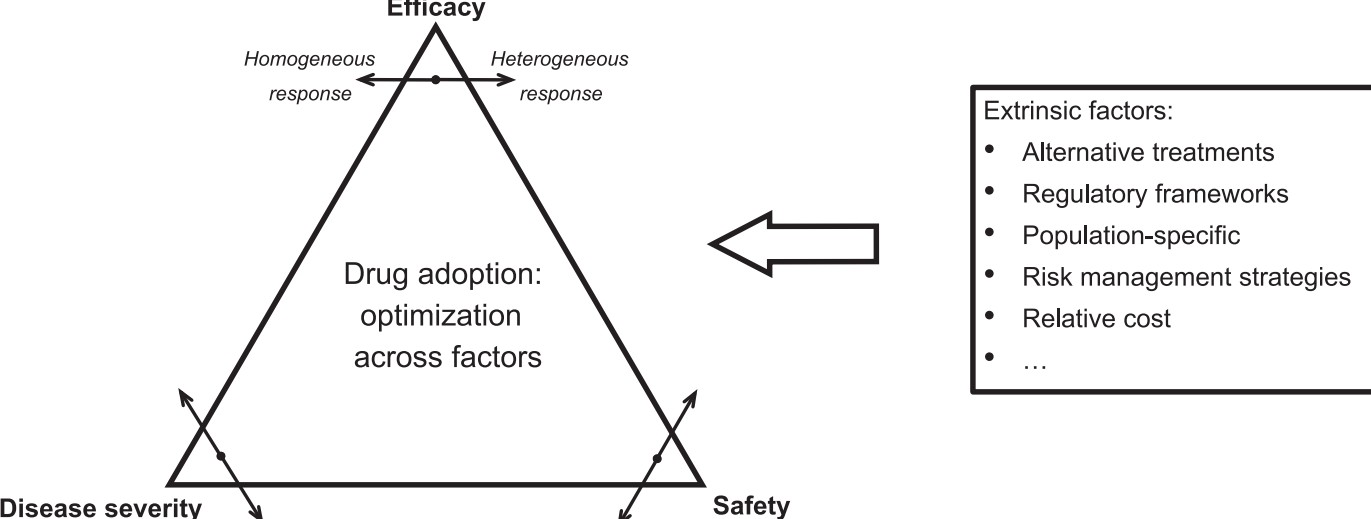

**Figure 8.  Factors affecting drug safety decision-making.**

necessitate a large number of observations (example drugs) to accurately profile their effects. One way to deal with this issue could be to mirror the drug development stages in separate steps of the computational screening pipeline. In the early stages, the drug development process primarily focuses on chemical features of screened compounds and their pharmacokinetic features, and then biomedical and clinical contexts are considered during laboratory and clinical trial testing. Computational models can be specialized to achieve optimal performance for each of these stages and combined to form a sequential filter, for example, approaches such as QED and ADME can be used as a first step, then tools such as PrOCTOR to identify compounds likely to fail during clinical trials, and lastly methods such as ours to identify the remaining problematic compounds. The only essential input required for TargeTox is the identity of proteins bound by a particular drug, whereas optional inputs, which may be missing, are the three Boolean values for possible routes of administration and two numeric values for lower and upper plasma protein binding. All of the other features, which are actually used by the trained model, are computed from the supplied list of bound proteins and the implementation released with this article can perform this part of the analysis automatically.

Our method is particularly dependent on the knowledge of proteins binding specific drugs, as these data are necessary to compute both the network-based and FI features. Idiosyncratic toxicity is often mediated by the effect on off-target proteins of particular drugs (57). Information about all possible bound proteins can often be incomplete, which can limit the effectiveness of the proposed method. Some resources offer computationally derived predictions of bound proteins (58), although use of such information would necessitate striking the right balance between true and false positives. Another aspect not currently considered by our model is the metabolism of the drug, which can generate toxic secondary compounds that may result in IT and can also have their own sets of protein interactions (59). The development of effective strategies for incorporating this wider body of knowledge can lead to further improvements of TargeTox and other similar methods.

Other means of making further progress could be in better utilization of other types of biological network data. This particular consideration was partially explored by considering distances derived from an integrated set of networks using the mashup method. Although in this particular case we have found that a single network of experimentally confirmed protein–protein interactions was marginally better, it is very likely that the better results could be obtained using different combinations of networks or different edge reliability thresholds. Although we have not been able to comprehensively explore all of these options as part of this initial study, we believe that a more in-depth evaluation of integrative methods such as mashup merits further investigation. In addition, incorporation of data from cell culture profiling studies offered by the Connectivity Map (60) and the more recent Library of Integrated Network-based Cellular Signatures (61) could be another way of more fully capturing the complexity of drug responses. The potential of combining such data with network-based approaches was recently demonstrated by the SynGeNet method (62), which successfully predicted genotype-specific drugs for melanoma.

In this work, we were particularly interested in exploring the problem of IT, that is, where the toxic effect is only manifested rarely

and therefore may be unnoticeable during clinical trials. We were able to confirm that the existing methods were not as effective in identifying these drugs. We have presented an example showing that our method can improve on the performance of existing methods specifically for those drugs. By applying TargeTox in a speculative way, we were also able to generate toxicity risk annotation for the druggable proteome. This follow-up analysis suggested that bound proteins associated with predicted toxicity risk are concentrated in highly specific areas of the human interactome and tend to have immune system and signaling-related functions. An additional insight was that the highest toxicity risk scores were only predicted by our model when a drug had several targets. This suggests that the burden of multiple drug–protein interactions on particularly susceptible regions of the networks could be a plausible hypothesis for explaining most severe cases of drug toxicity.

To facilitate applications of TargeTox, we have made the trained model, supporting data, and the necessary code available in a dedicated GitHub repository (https://github.com/artem-lysenko/TargeTox). Given that the only essential input for TargeTox is the identity of bound proteins for each drug, the method has particularly good synergy with the currently dominant target-driven paradigm of drug development. We believe that the method will be particularly useful for the identification of idiosyncratically toxic drugs during a computational screening of drug compounds and for the prior knowledge-directed design of combinations that minimize toxicity risk.

## Materials and Methods

### Dataset construction

The reference dataset was based on three resources: DrugBank (63) for drugs currently in use, ClinicalTrials.gov (64) for drugs that failed clinical trials, and supplementary data from (4) that compiled a comprehensive list of drugs that were withdrawn from the market between 1950 and 2016. The latter two resources were filtered manually to only keep the drugs that have failed for toxicity-related reasons. The DrugBank dataset was also filtered to remove all antineoplastic drugs, as those are expected to have relatively high toxicity to be considered sufficiently similar to the "safe" drugs for other diseases. To resolve any naming ambiguities, all drug names were mapped to ChEMBL identifiers using the DrugBank database and manual curation. Duplicates were removed to retain only one entry, with precedence given to the "toxic" class subset. Then, the ChEMBL database (65) was used to obtain the chemical structure information (SMILES strings) and bound proteins (both main pharmacological target(s) and any off-targets) for each drug. All entries where complete information was not available were discarded at this stage. This resulted in a set of 696 compounds in the "safe" category and 197 compounds in the "toxic" category. The ChEMBL and DrugBank databases were also used to obtain the pharmacological covariate data, specifically route of administration (oral, parenteral, and topical) and lower and upper estimates for blood plasma protein binding, although missing values were allowed for these features.

Data for proteins bound by each drug were integrated with a protein association network from the STRING database (66). To

control for false-positive edges, we only used experimentally confirmed interactions with a combined score of at least 200. To ensure consistent distribution of distances, only the main connected component of this network containing 16,610 proteins was used for all of the analyses. Information about the biological function of the bound proteins was acquired from the Gene Ontology [67] annotation database and annotation of drugs with side effects—from the OFFSIDES database [39]. Additional literature curation was performed to identify a subset of drugs with reported IT, which is provided in supplementary materials (Table S3). Drugs associated with HLA-mediated toxicity were identified based on the list from [37] and are, likewise, provided in Table S4.

### Computation of candidate-predictive features

Chemical structure was used to compute drug properties using the ChemmineR package [68] applied as specified in PrOCTOR analysis script [10]. In addition, we ran all the drugs in our dataset through PrOCTOR to obtain the PrOCTOR score and weighted QED (wQED) [9].

Several different network distance metrics were evaluated for inclusion in our model. The simpler metrics considered were the shortest path length in STRING protein–protein interaction graph and the discretized shortest path where a value 1 was assigned if the length was less than 3 and 0 otherwise. Of the more advanced measures, we have considered mashup, which computes the distance over an integrated set of multiple biological networks, and the DSD algorithm, a distance measure based on random walks. In the case of Mashup, we used the pre-computed matrix of vectors for STRING networks made available by the authors of the algorithm. The matrix was transformed to a distance matrix by computing the cosine distances between all pairs of vectors. Cosine distance was chosen because it was suggested as the most appropriate one in the original mashup method article.

In the latter case, the network was transformed into a symmetrical DSD as described in the work of [33], using the following formula:

$$\mathrm{DSD}(u, v) = \left\| \left( b_u^T - b_v^T \right) \left( I - D^{-1}A + P \right)^{-1} \right\|_1,$$

where $D$ is the diagonal degree matrix, $P$ is the constant matrix of the steady-state distribution and $b_u$, $b_v$ are the basis vectors for the respective nodes. The DSD metric allows the fine-grained mapping of all drug-binding proteins into a network-predicated topological space. Using this distance metric, we were able to compare the relative distributions of different bound proteins sets. As we found that proteins that bind to the same drug tended to be located close to each other in the network, the position of the set can be approximated by the position of its convex hull with respect to a few reference nodes. This transformation summarizes the biological network location of any set of possible bound proteins in the same small number of variables—regardless of its original size.

Next, we explored several possible designs for the network-based features. The options considered were representing each drug by a network-based medoid for all bound proteins of a particular drug and using a full set of distances between closest bound protein and each other node in the network. Given that the

latter most promising strategy had considerable computational costs, we explored how the number of reference points could be reduced. Specifically, the reference nodes were chosen to be the most representative with respect to the set of all drug-binding proteins, with the rationale being that this proximity will serve to reduce possible noise and errors due to unavoidable missing or spurious edges in the protein association network. Candidate reference nodes were identified by computing enrichment for drug binding proteins in a fixed-distance neighborhood around each node in the network (Fig 9). To reduce redundancy, all significantly enriched neighborhoods were clustered using hierarchical clustering to group them into the desired number of distinctive groups. Finally, the representative central node of the densest neighborhood in each cluster was chosen as a reference node. Three free parameters of this approach (neighborhood-defining distance, enrichment cut-off and number of clusters) were optimized using grid search.

FI score metric was derived from the Gene Ontology BP annotations. For the purposes of this analysis annotations to each term also inherit annotation to all of its ancestor terms. Using a complete set of all human annotations, information content was computed for each of the individual terms:

$$\mathrm{IC}(t) = -\ln\left( \frac{|k_t|}{|k|} \right)$$

where $t$ is a given GO term and $k$ is an instance of annotation (unique entity–term pair). Then, FI score is defined as follows:

$$\mathrm{FI}(T) = \sum_{t_i \in T} \mathrm{IC}(t_i)[\mathrm{descendants}(t_i) \cap T = \varnothing],$$

where $T$ is a set of all annotation terms for a set of particular drug-binding proteins. The rationale behind the design of this feature is

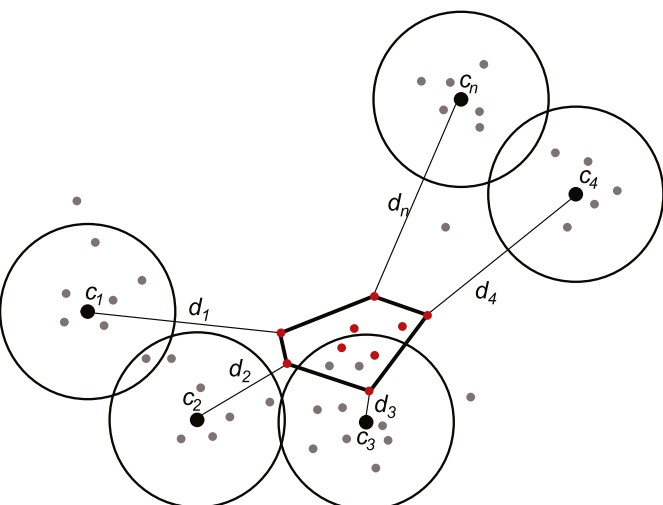

**Figure 9. Conceptual schematic of the network-based feature design.**
All protein nodes are mapped represented by their DSD vectors (filled dots). A set of reference nodes ($c_1$–$c_n$; black dots) are chosen in the areas with dense concentration of known drug targets by computing enrichment in a fixed-distance area around each candidate node. The redundancy is then removed using hierarchical clustering of all qualifying candidates. The network distance-based features are defined as a distance ($d$) between a reference node and the closest target for a given drug (red dots).

that a drug is expected to have an impact on some BPs as part of its intended mechanism of action. If this impact is focused, there will be few other processes affected, so the score will be relatively low. On the other hand, if a drug also affects some off-target processes or interacts with a critical protein contributing to multiple processes, the FI score will be high. Information content serves to achieve even further granularity of the measure, as it is low for generic functions that are relatively common and high for specialized functions where there is little redundancy.

### Classifier training and evaluation

Here, we describe basic notations for training and evaluating our proposed model. Let $\chi$ be a set of $n$ samples in a $d$-dimensional feature space which is split into a training set $\chi_{tr}$ and test set $\chi_{ts}$; that is, $\chi = \chi_{tr} \cup \chi_{ts}$. Let $\Omega = \{\omega_i : i = 1, 2, ...c\}$ be the finite set of $c$ class labels, and $\omega_i$ is the class label of $i$th class. To preserve the classes, the training and test sets are subdivided into $c$ disjoint subsets $\chi_{tr} = \chi_{tr1} \cup \chi_{tr2} ... \cup \chi_{trc}$ and $\chi_{ts} = \chi_{ts1} \cup \chi_{ts2} ... \cup \chi_{tsc}$, respectively, where $\chi_{trj} \subset \chi_{tr}$ and $\chi_{tsj} \subset \chi_{ts}$. Furthermore, it can be noted that each subset $\chi_{trj}$ or $\chi_{tsj}$ has class a label $\omega_j$. Let $n_{trj}$ and $n_{tsj}$ be the number of samples in $\chi_{trj}$ and $\chi_{tsj}$ (of class $\omega_j$) such that

$$n_{tr} = \sum_{j=1}^{c} n_{trj}$$

and

$$n_{ts} = \sum_{j=1}^{c} n_{tsj}$$

The feature vectors of $\chi_{tr}$ and $\chi_{ts}$ can be depicted as

$$\chi_{tr} = \{r_1, r_2, ...r_{ntr}\}$$

and

$$\chi_{ts} = \{s_1, s_2, ..., s_{nts}\}.$$

To perform a robust evaluation of the model, we split up our data into a training subset $\chi_{tr}$ and validation subset $\chi_{ts}$ in $n_{tr}:n_{ts} = 80:20$ ratio while preserving the ratio of the two classes in our dataset; that is, $n_{tr1}/n_{tr2} \approx n_{ts1}/n_{ts2}$. During development, evaluation was performed using a five-fold cross-validation approach using only the compounds in the training set and the final evaluation was performed using the hold-out set.

To train our model, we have chosen to use a gradient-boosting algorithm. The objective of the gradient boosting algorithm is to find an approximation $\widehat{F}(x)$ of a function $F(x)$ such that the expected value of a loss function $L(\cdot)$ is minimum (69); that is,

$$\widehat{F} = \text{argmin}_F E[L(y, F(x))],$$

where $y$ is the class label of a feature vector $x$, and $E[\cdot]$ is the expectation function. Gradient boosting is generally used with decision trees $h(x)$. The $t$-th step gradient boosting with decision trees $h_t(x)$ is updated in Friedman's algorithm as follows:

$$\widehat{F}_t \leftarrow \widehat{F}_{t-1} + \gamma_t h_t(x),$$

where the step size $\gamma_t$ is selected such that the loss function $L(\cdot)$ is minimized. For this work, we have used a gradient-boosting tree classifier ensemble implementation from the "catboost" v0.10.3 R

library (70 *Preprint*). Classifier hyper-parameters and parameters of the network feature design were tuned on the test set using grid search, and then the optimal configuration was validated on the hold-out set and used to train the final model. Primary evaluation of performance was performed on the basis of area under the ROC AUC, computed using "PRROC" R package with "toxic" class instances set as the foreground class.

### Feature importance analysis

Feature importance analysis was performed using implementations available in the "catboost" R library, which allows computation of canonical the decision tree ensemble importance scores and SHAP score metrics. Importance scores were computed for the final model that was trained on the complete dataset (i.e., a union of train and test subsets). The SHAP scores were computed for each individual drug by running a leave-one-out cross-validation on an entire set. Two types of comparisons were performed, which aimed to profile the differences between the "idiosyncratically toxic" and "clinical trial toxic" subsets. The first looked at the averages of all per-feature SHAP scores and compared their distributions using Wilcoxon signed-ranks test. The second comparison only considered feature scores that contributed to the correct classification of respective drugs as toxic and compared the relative totals allocated to each feature in percentage terms.

### Validation using side-effects data

Side-effects information was downloaded from the OFFSIDES database (39). This drug annotation was combined with the STITCH database (58) to map them to the protein association network. As opposed to ChEMBL, which was used to construct our training data, STITCH also includes speculative protein-binding annotation. Therefore, to make these datasets comparable, only high-confidence (score > 800) annotations and proteins also present in the protein–protein association network were retained. The drugs were filtered to remove those present in either the training or hold-out subsets, which resulted in 339 compounds being retained. From the set of available annotations for those drugs, we have selected all major toxicity-associated side effects with at least 10 occurrences, which resulted in 14 categories. These side effects included most categories commonly linked to drug withdrawals (4), such as cardiotoxicity, hepatotoxicity, and nephropathy. The predicted scores of drugs in those categories were compared with the remaining subset which did not have any of these annotations using Wilcoxon signed-ranks test.

### Model interpretation and annotation of the druggable proteome

The overall contribution of individual features to the model was quantified with a feature importance metric and possible relationships between individual features with an interaction strength metric using implementations included in the "catboost" library. To extract the overall map of toxicity risk from the model, we used a druggable genome dataset from (40). The reasoning behind this choice was that these drug-binding proteins are both most relevant and most likely to be consistent with the data used for training. For

each protein, a simulated parenteral drug instance was generated using real DSD distance to the reference points and GO functional annotation of that protein, with remaining features set to missing. To explore possible patterns, distances of these proteins to 12 reference points were mapped onto a 2-D space using the t-SNE algorithm with default settings. To further profile areas of the highest predicted toxicity, the top 10% of proteins by score were analyzed as a separate set. Specifically, modules in this subset were identified by fitting a Gaussian finite mixture model using the expectation-maximization algorithm. This analysis was performed using an implementation from "pvclust" R package (71). Then, functional enrichment test was performed for each identified module using Fisher's exact test followed by Benjamini–Hochberg false discovery rate correction.

An implementation of the method and the supporting data have been made available in a public GitHub repository with the following URL: https://github.com/artem-lysenko/TargeTox. All other data used in this study were acquired from the relevant public resources as identified in the Materials and Methods section.

## Supplementary Information

## Acknowledgements

This work was supported by Core Research for Evolutional Science and Technology grant from the Japan Science and Technology Agency (grant no. JPMJCR1412), and Japan Society for the Promotion of Science KAKENHI (grant nos 18K18156, 17H06307, and 17H06299).

### Author Contributions

A Lysenko: conceptualization, software, formal analysis, validation, investigation, visualization, methodology, and writing—original draft, review, and editing.
A Sharma: methodology and writing—review and editing.
K Boroevich: resources, data curation, software, and writing—review and editing.
T Tsunoda: conceptualization, supervision, funding acquisition, project administration, and writing—review and editing.

### Conflict of Interest Statement

The authors declare that they have no conflict of interest.

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
