## [Reviewer comments · Life Science Alliance]

Life Science Alliance

An integrative machine learning approach for prediction of toxicity-related drug safety

Artem Lysenko, Alok Sharma, Keith Boroevich, and Tatsuhiko Tsunoda

DOI: [10.26508/lsa.201800098](https://doi.org/10.26508/lsa.201800098)

Corresponding author(s): Artem Lysenko, RIKEN and Tatsuhiko Tsunoda, RIKEN Center for Integrative Medical Sciences

Review Timeline:

Submission Date:	2018-05-30
Editorial Decision:	2018-06-26
Revision Received:	2018-10-27
Editorial Decision:	2018-11-11
Revision Received:	2018-11-20
Accepted:	2018-11-20

Scientific Editor: Andrea Leibfried

Transaction Report:

June 27, 2018

Re: Life Science Alliance manuscript #LSA-2018-00098-T

Dr. Artem Lysenko
RIKEN
Center for Integrative Medical Sciences
1-7-22 Suehiro-cho
Tsurumi
Yokohama, Kanagawa 230-0045
Japan

Dear Dr. Lysenko,

Thank you for submitting your manuscript entitled "Integrative analysis of protein association networks for prediction of drug toxicity" to Life Science Alliance. The manuscript was assessed by expert reviewers, whose comments are appended to this letter.

As you will see, the reviewers think that your method could be of value to others. However, they also note some issues that need to get addressed to allow publication in Life Science Alliance. Importantly, both reviewer #1 and #3 criticise the way the performance of TargeTox was compared to the one of PROCTOR and QED. It would be crucial in our view to evaluate TargeTox's performance in a satisfactory way. All three reviewers provide other input that should get addressed to improve the experimental setup and methodology and to thus strengthen the value of TargeTox for others. We would therefore like to invite you to submit a revised version of your work, addressing the reviewers' concerns. Please note that we will need strong support from reviewer #1 and #3 on such a revised version.

- A letter addressing the reviewers' comments point by point.
- An editable version of the final text (.DOC or .DOCX) is needed for copyediting (no PDFs).
- High-resolution figure, supplementary figure and video files uploaded as individual files: See our detailed guidelines for preparing your production-ready images, <http://life-science-alliance.org/authorguide>

B. MANUSCRIPT ORGANIZATION AND FORMATTING:

Full guidelines are available on our Instructions for Authors page, <http://life-science-alliance.org/authorguide>

Thank you for this interesting contribution to Life Science Alliance. We are looking forward to receiving your revised manuscript.

Sincerely,

Reviewer #1 (Comments to the Authors (Required)):

How to reliably predict drug toxicity from in silico aspect is one of important links to make a better drug. Lysenko et al present TargeTox, a machine learning-based method, namely using gradient boosted decision tree ensemble algorithm to assess drug toxicity via distance-based network features. TargeTox combines protein network analyses with chemical analyses developed by others in previous work to score drugs based on a model developed by machine learning classifier. The model was evaluated on the remaining subset of Onakpoya et al. and drugs from the OFFSIDES database. Overall the TargeTox technique was conceived to address an area of great need and promise for computational techniques, the computational results are encouraging to predict drug toxicity, in particular idiosyncratic toxicity. However, there are a few major concerns to be addressed before the manuscript is deemed publishable.

Major concerns:

1. The key purpose of this study is to design a robust algorithm that can predict idiosyncratic toxicity of drugs. However, the logic and rationale of the choice of data use to train the model and algorithm chosen (in this case gradient boosted decision tree ensemble algorithm) are poorly stated. In particular, why the choice of such training data and algorithm enable one to distinguish idiosyncratic from "non-idiosyncratic" toxicities?
2. Idiosyncratic toxicities (also called type B reactions) are toxicities of drugs that rarely and occur unpredictable amongst the population. Within the manuscript, the authors describe idiosyncratic toxicity as "...idiosyncratic toxicity cases (list of drugs withdrawn from the market due to high toxicity)" in Introduction, "As this list is composed of drugs withdrawn from the market due to unacceptable toxicity, it is enriched for idiosyncratic toxicity side effects that were not discovered in smaller populations during clinical trials" in Result section, and "... idiosyncratic toxicity, i.e. where the toxic effect is only manifested rarely and therefore may be missed during clinical trials" in Discussion. What I don't understand in this work is, the authors stated the list of drugs they used for training was "enriched for idiosyncratic toxicity side effects" because the data of drug list "is composed of drugs withdrawn from the market due to unacceptable toxicity". Based on what criteria the authors ensure these drugs are enriched with idiosyncratic toxicities that is "unpredictable" in clinic? Otherwise, the findings made from this work might be confounded with features due to high and frequently occur toxicities and were of course rejected because of unacceptable toxicity, as the authors also stated.
3. In Figure 1, the authors compared the performance of ProCTOR and QED and illustrated their poor predictive power to the so-called idiosyncratic toxicities. In my opinion, this is unfair comparison. First, the data used to build these two models are different from this work and the aim of building such models are not for predicting idiosyncratic toxicities per se. The way the authors present can easily led to misunderstanding that current existing methods such as ProCTOR and QED are poorly performed. The authors should rephrase their statements in order to avoid such confusion. Otherwise, Figure 1 should be removed from the manuscript. Second, the authors have to justify what features are indeed associated with idiosyncratic toxicities in order to claim current method is indeed more superior in predicting idiosyncratic toxicity that current existing methods are lacking. I believe if the authors can provide such justification, this will add large merit to the current work!
4. Idiosyncratic toxicities- One of the main stated motivations was the lack of methods that could

accurately identify idiosyncratic toxicities. This aim was not well addressed by the figures shown. The Onakpoya database was enriched in drugs with idiosyncratic toxicities, but the analysis only showed that TargeTox improved (by measure of ROC AUC in both subsets Fig3B and C) on previous methods. A figure specifically addressing TargeTox's ability to address idiosyncratic toxicities is needed because of the importance of these toxicities in the drug development process. Including ProCTOR and QED in this figure would further help differentiate TargeTox's ability to excel at identifying idiosyncratic toxicities.

5. In the Introduction, the author stated "The sparse and complex structure of these data makes it challenging to use with modern machine learning methods, which usually require dense and regular data as input". Is this true? My understanding is this is precisely the power of machine learning methods, in particular Support Vector Machine (SVM) and Artificial Neural Network (ANN) to extract hidden features from high-dimensional and heterogeneous data. Also, in the Introduction, the author also claims "In our approach, this is solved by mapping targets onto a biological network, which is then transformed into a set of points in diffusion state distance (DSD) space". To me, this is over claimed and as mentioned, SVM and ANN are capable to deal with such data complexity, the authors should clarify these statements carefully.

6. The data used to build the model consisted 696 safe and 123 toxic drugs and the authors had excluded "negative" examples from the ClinicalTrials.gov database. Isn't that including such negative examples might improve model performance (considering the unbalanced dataset)? The authors need to justify this.

Minor concerns:

1. Statement of needed inputs- If my understanding is correct, drug-protein interactions, chemical structure, and plasma binding data is the only needed inputs to score a drug with TargeTox. Mention of these inputs (and others if there are any) all in one place would be helpful in the discussion of where TargeTox can be implemented in the drug development pipeline.

2. In Figure 3A there is some overlapping text below the training set box

3. In Figure 2 the properties of drugs and their targets are analyzed for those contained in DrugBank and ChEMBL. Do these trends still apply for the Onakpoya dataset? Repeating this analysis for the Onakpoya is recommended as most of the paper focuses on the Onakpoya dataset.

4. What was the rationale for using a gradient boost decision tree ensemble over other methods?

5. The abstract mentions that TargeTox incorporates drug target biological function. Where within TargeTox is this incorporated?

Reviewer #2 (Comments to the Authors (Required)):

The proposed study is well designed (data collection, filtering, can be used as a resource) and well evaluated (compared with other methods) with good results. Some comments:

In addition to the known targets, the unknown/off-targets of drugs may play important roles causing

the toxic effects. As an additional independent data resource, the Connectivity Map (CMap) (<https://www.ncbi.nlm.nih.gov/pubmed/17008526>) might be helpful. Integrating drug targets and CMAP has been used in cancer drug prediction (e.g., <https://www.ncbi.nlm.nih.gov/pmc/articles/PMC5543336/>). This should be added into discussion.

Minor: figure resolution is low (not clear)

Reviewer #3 (Comments to the Authors (Required)):

Lysenko et al. describe a network-based approach (TargeTox) for identifying drugs that will likely be withdrawn from the market for toxicity reasons. The approach takes a protein-protein interaction (PPI) network and uses diffusion state distance, a metric based on graph diffusion property, to calculate distances between every pair of proteins taking into account their broader neighborhood in the PPI network. Using this distance metric, Lysenko et al. then derive a feature representation for every drug in the database. This is achieved by taking proteins targeted by the drug and summarizing their corresponding points in the diffusion state space by a convex hull. The resulting feature representations of drugs are used to train a binary classifier (gradient boosting trees) to predict the likelihood that a given drug will be safe for patients

In a cross-validation study, the new approach achieves an AUROC of 72%. Additionally, Lysenko et al. perform an independent validation study using drug side-effect information and show that predicted toxicity scores are higher for drugs that have stronger toxicity-related side effects.

The focus on rare idiosyncratic effects is interesting and novel, because these are very difficult to predict and the mechanism is often immunological-the authors might want to discuss their ability to predict HLA-mediated hypersensitivity syndromes specifically (e.g. for abacavir, carbamazepine, phenytoin etc...)

The other interesting component is the findings that drug side effects are related to the functional impact score, and that "fake" drugs that only have one target have a much more restricted set of side effects. This is consistent with the idea that drug actually have pleiomorphic effects on physiology, and not just primary target-directed effects.

This work addresses an important problem in modeling drug toxicity and is potentially interesting. However, I have several concerns related to experimental setup and methodology that I describe below.

1. Lysenko et al. use diffusion state distance (DSD) to learn distances between proteins in the PPI network. DSD method was published in 2013 and has since been extended and improved in several different ways. For example, Mashup (Cho et al., Cell Systems 2016) is one such method that outperforms DSD by 26% when applied to the same PPI network as used in this paper. It would be interesting to see whether these recent methodological advancements can further improve TargeTox.

2. TargeTox approach is not evaluated against any baseline method. Because of that, it is difficult to judge how good an AUROC value of 72% is for this prediction task (Figure 3BC). While Lysenko et al. report performance of two existing methods for drug toxicity prediction (i.e., ProCTOR and QED) in Figure 1, the values in the figure do not seem to be directly comparable to the values in Figure 3BC. Can authors clarify this issue? When comparing different methods with each other, one

needs to use the same experimental setup and the same set of testing drugs.

3. It would be interesting to better understand what components of TargeTox are most important for its good performance. Ideally, one would implement a series of increasingly strong baselines and compare them to full TargeTox. This paper raises several basic questions that remain unanswered. First, how does the performance of TargeTox change if one replaces DSD with a simple baseline based on the shortest distance between proteins in the PPI network or the presence of connecting paths of length two between proteins? Second, how does the performance of TargeTox change if one calculates a drug's feature representation by aggregating proteins' representations via simple averaging instead of a convex hull? Third, how important is it to compute a convex hull considering only a few reference points instead of the entire set of drug targets? None of these questions are analyzed computationally and answered in the paper.

In addition, the author use two methods ProCTOR and QED to show that their methods are better. In the last paragraph of "Introduction", they mentioned that "To our knowledge, of the currently existing integrative methods, only ProCTOR satisfies both of these criteria, as it was specifically developed to predict failure of clinical trials. However, one of the criteria as they mention is " have no reliance on the types of information not readily available during drug development process, like drug response human gene expression data ". I am not convinced by the way they reason that ProCTOR is the only method that they want to compare with.

Their new method uses drug target information. They argued that the issue of data sparsity and complexity can be solved by mapping drug targets onto a biological network by transforming to DSD space. I searched the whole paper, how this can be solved, theoretically, was not discussed. Drug target information was obtained from DrugBank. They should have at least mentioned/discussed that the effects from hidden off-targets.

I understand that they used network context as input candidate features for classifiers, but it is still confusing to read figure labels in Figure 3E and Figure 3F.

In section of "model interpretation", authors discussed their predicted "toxicity risk map". From their description of the clusters of "toxicity risk map", it looks like these genes are playing very important roles in biological functions. It would be interesting to further investigate if these genes are already known to be associated with drug adverse reactions. We suggest authors to address this.

We would like to thank all of the reviewers for considering our paper and suggesting ways to improve it further. As well as performing additional analysis requested we have updated all of the analysis and code released with this paper to use the latest version of the Catboost library, which lead to slightly better results and some slight differences with the previous version of the manuscript, though did not lead to major changes in any conclusions or interpretation. Additionally we have made one correction, where we found that druggable proteome annotation in the previous version used model was not trained on the complete dataset. This change somewhat clarified the pattern reported from that part of the analysis.

Our response to individual points is as follows.

Reviewer 1

Major

1. The key purpose of this study is to design a robust algorithm that can predict idiosyncratic toxicity of drugs. However, the logic and rationale of the choice of data use to train the model and algorithm chosen (in this case gradient boosted decision tree ensemble algorithm) are poorly stated. In particular, why the choice of such training data and algorithm enable one to distinguish idiosyncratic from "non-idiosyncratic" toxicities?

We acknowledge that original version of the manuscript had placed too much emphasis on detection of idiosyncratic toxicity and not enough analysis/justification was done to support this. We have added a number of references to support the claim that idiosyncratic toxicity is the lead cause of drug withdrawals from market, which would mean that our method can potentially capture some of these effects because more of these examples are likely to be present in such data. Additional analysis was done to explore this further as was suggested by the reviewer in another point. From the more general methodological perspective, our approach is primarily concerned with development of possible ways of using biological network data to further enhance biomedical machine learning analysis. Therefore other modern algorithms like SVMs or neural networks can be used with our method instead of GBMs. We have added text to highlight this point and explain our reason for choosing a GBM. And, lastly, we have included a statement to clarify that although our method may predict potential for both idiosyncratic and non-idiosyncratic toxicities in current form it does not directly provide means to distinguish these two sub-categories from each other.

2. Idiosyncratic toxicities (also called type B reactions) are toxicities of drugs that rarely and occur unpredictable amongst the population. Within the manuscript, the authors describe idiosyncratic toxicity as "...idiosyncratic toxicity cases (list of drugs withdrawn from the market due to high toxicity)" in Introduction, "As this list is composed of drugs withdrawn from the market due to unacceptable toxicity, it is enriched for idiosyncratic toxicity side effects that were not discovered in smaller populations during clinical trials" in Result section, and "... idiosyncratic toxicity, i.e. where the toxic effect is only manifested rarely and therefore may be missed during clinical trials" in Discussion. What I don't understand in this work is, the authors stated the list of drugs they used for training was "enriched for idiosyncratic toxicity side effects" because the data of drug list "is composed of drugs withdrawn from the market due to unacceptable toxicity". Based on what criteria the authors ensure these drugs are enriched with idiosyncratic toxicities that is "unpredictable" in clinic? Otherwise, the findings made from this work might be confounded with features due to high and frequently occur toxicities and were of course rejected because of unacceptable toxicity, as the authors also stated.

We have added several extra references to support the statement that idiosyncratic toxicities are the main toxicity-related cause of market withdrawals. We have searched the literature to identify which drugs in our dataset were identified to be idiosyncratically toxic and done analysis to show that our method can distinguish them from the 'safe' subset (Fig 3E). We have made changes to the text to more closely align our claims and results of this additional analysis and removed or rephrased the problematic statements identified by the reviewer.

3. In Figure 1, the authors compared the performance of ProCTOR and QED and illustrated their poor predictive power to the so-called idiosyncratic toxicities. In my opinion, this is unfair comparison. First, the data used to build these two models are different from this work and the aim of building such models are not for predicting idiosyncratic toxicities per se. The way the authors present can easily lead to misunderstanding that current existing methods such as ProCTOR and QED are poorly performed. The authors should rephrase their statements in order to avoid such confusion. Otherwise, Figure 1 should be removed from the manuscript. Second, the authors have to justify what features are indeed associated with idiosyncratic toxicities in order to claim current method is indeed more superior in predicting idiosyncratic toxicity that current existing methods are lacking. I believe if the authors can provide such justification, this will add large merit to the current work!

We agree with the concerns raised by the reviewer. The figure in question has been moved to the supplementary and we replaced this text with more accurate version that highlights the points made by the reviewer. In the main text we have emphasized our view that complexity of drug development requires use of multiple methods to cover all possible aspects. We do not claim that our approach can completely supersede other methods, but merely fill in important gaps in current capabilities, e.g. facilitate identification of drugs which can successfully pass clinical trials but turn out to be toxic after release to market or cases when understanding of possible effect of binding specific proteins is of interest. To identify which of our features are associated with idiosyncratically toxic drug examples we have analyzed and compared distributions of Shapley values from our model for these different toxicity sub-types. This analysis indicated that there are indeed differences specific to the idiosyncratic subset and, in particular, differences were significant for several of the network-based features (Table 1).

4. Idiosyncratic toxicities- One of the main stated motivations was the lack of methods that could accurately identify idiosyncratic toxicities. This aim was not well addressed by the figures shown. The Onakpoya database was enriched in drugs with idiosyncratic toxicities, but the analysis only showed that TargeTox improved (by measure of ROC AUC in both subsets Fig3B and C) on previous methods. A figure specifically addressing TargeTox's ability to address idiosyncratic toxicities is needed because of the importance of these toxicities in the drug development process. Including ProCTOR and QED in this figure would further help differentiate TargeTox's ability to excel at identifying idiosyncratic toxicities.

We have added the requested figure to the supplementary, including results for the QED and ProCTOR produced on the same set of drugs (Fig S3).

5. In the Introduction, the author stated "The sparse and complex structure of these data makes it challenging to use with modern machine learning methods, which usually require dense and regular data as input". Is this true? My understanding is this is precisely the power of machine learning methods, in particular Support Vector Machine (SVM) and Artificial Neural Network (ANN) to extract hidden features from high-dimensional and heterogeneous data. Also, in the Introduction, the author also claims "In our approach, this is solved by mapping targets onto a biological network, which is then transformed into a set of points in diffusion state distance (DSD) space". To me, this is over

claimed and as mentioned, SVM and ANN are capable to deal with such data complexity, the authors should clarify these statements carefully.

We have rephrased that section to be more specific about how our method can counter the data sparseness problem specifically, added a statement saying that our method can be used with other types of machine learning methods and explained for what reasons GBM was chosen.

6. The data used to build the model consisted 696 safe and 123 toxic drugs and the authors had excluded "negative" examples from the ClinicalTrials.gov database. Isn't that including such negative examples might improve model performance (considering the unbalanced dataset)? The authors need to justify this.

We have corrected the statement to be more precise about this point. The drugs removed were anti-cancer drugs, which were classed as 'safe' according to our original criteria (i.e. no clinical trials failed and no market withdrawals specifically due to toxicity). Anti-cancer drugs can be considered a special case where some examples of highly toxic compounds are expected even though the drug remains in use. We have already explored the possibility of adding those as 'toxic' class during method development, however found that it lead to decrease in performance, most likely due to the set having both highly toxic and well-targeted drugs with lower risks. This explanation was added to the text.

Minor

1. Statement of needed inputs- If my understanding is correct, drug-protein interactions, chemical structure, and plasma binding data is the only needed inputs to score a drug with TargeTox. Mention of these inputs (and others if there are any) all in one place would be helpful in the discussion of where TargeTox can be implemented in the drug development pipeline.

We have added this list of inputs at the location suggested by the reviewer.

2. In Figure 3A there is some overlapping text below the training set box

The issue is now fixed.

3. In Figure 2 the properties of drugs and their targets are analyzed for those contained in DrugBank and ChEMBL. Do these trends still apply for the Onakpoya dataset? Repeating this analysis for the Onakpoya is recommended as most of the paper focuses on the Onakpoya dataset.

We have done this analysis for the Onakpoya dataset and added a new figure (Fig 2B) to show these results.

4. What was the rational for using a gradient boost decision tree ensemble over other methods?

We have added the following explanation for our choice of the algorithm to the text: "... In principle, this strategy can be used in combination with any modern classifier that has some form of regularization capabilities and can handle non-linear relationships, e.g. certain SVM variants or deep neural networks. However, in this case gradient boosted classifier tree ensemble model (GBM) was chosen for the following two reasons. First was the small numbers of positive (toxic) drugs in our training dataset, which meant that comparatively less hyper-parameter tuning required by GBM was considered to be very helpful for mitigation of the overfitting risk. Second reason was the presence of missing values in our data, which GBM can handle without the need for prior imputation, thereby greatly simplifying both development and any possible future applications of our method."

5. The abstract mentions that TargeTox incorporates drug target biological function. Where within TargeTox is this incorporated?

Biological function data is included in the form of Gene Ontology annotation, which is used to compute the functional impact score feature, as outlined in the Methods section. We have edited the statement in question to be clearer on this point.

Reviewer 2

Major

In addition to the known targets, the unknown/off-targets of drugs may play important roles causing the toxic effects. As an additional independent data resource, the Connectivity Map (CMap) (<https://www.ncbi.nlm.nih.gov/pubmed/17008526>) might be helpful. Integrating drug targets and CMAP has been used in cancer drug prediction (e.g., <https://www.ncbi.nlm.nih.gov/pmc/articles/PMC5543336/>). This should be added into discussion.

We have improved our discussion of particular importance of off-target effects and highlighted the relevant works suggested by the reviewer.

Minor

figure resolution is low (not clear)

We included high-quality figures with this submission of the manuscript.

Reviewer 3

The focus on rare idiosyncratic effects is interesting and novel, because these are very difficult to predict and the mechanism is often immunological-the authors might want to discuss their ability to predict HLA-mediated hypersensitivity syndromes specifically (e.g. for abacavir, carbamazepine, phenytoin etc...)

We have explored the potential of our model to predict the HLA-mediated toxicity drugs that were present in our dataset as was suggested by the reviewer. One issue we have found is that most of the relevant drugs were classed as 'safe' according to our chosen criteria (no clinical trials failed and no market withdrawals specifically due to toxicity). We believe this is likely due to availability of strategies to manage the risks because this mechanism is now well-researched (e.g. knowledge of specific populations / allele carriers who could be at risk). The results of this analysis are now included in the paper (Fig 3F).

1. Lysenko et al. use diffusion state distance (DSD) to learn distances between proteins in the PPI network. DSD method was published in 2013 and has since been extended and improved in several different ways. For example, Mashup (Cho et al., Cell Systems 2016) is one such method that outperforms DSD by 26% when applied to the same PPI network as used in this paper. It would be interesting to see whether these recent methodological advancements can further improve TargeTox. We have done the comparison with Mashup and included these results in the paper (Fig S2). Although DSD based implementation was still marginally better in the discussion we have highlighted that Mashup and similar approaches offers much more possibility for customization (e.g. integration of different networks) which has great potential for future development of this and similar methods.

2. TargeTox approach is not evaluated against any baseline method. Because of that, it is difficult to judge how good an AUROC value of 72% is for this prediction task (Figure 3BC). While Lysenko et al. report performance of two existing methods for drug toxicity prediction (i.e., ProCTOR and QED) in

Figure 1, the values in the figure do not seem to be directly comparable to the values in Figure 3BC. Can authors clarify this issue? When comparing different methods with each other, one needs to use the same experimental setup and the same set of testing drugs.

This was done in this way because pre-trained model of ProCTOR provided by the authors already used all of the examples from ClinicalTrials.gov, which would bias the results if they were included in the evaluation set. We agree that ideally exact comparison would have been interesting and have already considered this possibility during model development. From inspecting the ProCTOR model, we could see that it had used a balanced dataset of 100 'toxic' and 100 'safe' drugs, however the paper did not provide sufficient details about their identity or how these 200 drugs were chosen from a much larger number available. This meant that we could not re-train the random forest classifier of the ProCTOR model to be methodologically comparable. An additional complication, which was also highlighted by the reviewer 1, is that features selected for ProCTOR method were chosen for a different goal (prediction of clinical trial success) and optimized on a different dataset (our dataset included substantial number of drugs that were found to be toxic after passing the clinical trials). Therefore, we aimed to emphasize the complementary of our approach by illustrating that ProCTOR/wQED do not perform as well for a specific set of drugs (idiosyncratically toxic drugs). For this specific case we have now added a new figure (Fig S3) to demonstrate this point using the same set of drugs for all of these methods.

3. It would be interesting to better understand what components of TargeTox are most important for its good performance. Ideally, one would implement a series of increasingly strong baselines and compare them to full TargeTox. This paper raises several basic questions that remain unanswered.

We have implemented all of the baselines suggested by the reviewer and done the comparisons to address this. We have also done additional feature attribution analysis using SHAP metrics in addition to importance analysis, which elaborates this point further.

3a First, how does the performance of TargeTox change if one replaces DSD with a simple baseline based on the shortest distance between proteins in the PPI network or the presence of connecting paths of length two between proteins?

We have added this comparison to the paper.

3b Second, how does the performance of TargeTox change if one calculates a drug's feature representation by aggregating proteins' representations via simple averaging instead of a convex hull?

We have added this comparison to the paper, with one small modification: as DSD is distance rather than coordinate matrix we have used a medoid instead of an average.

3c Third, how important is it to compute a convex hull considering only a few reference points instead of the entire set of drug targets? None of these questions are analyzed computationally and answered in the paper.

We have explored the effect of using larger number of reference points and included results in the paper. The effect is largely accounted for by the feature selection and regularizing inherent to the GBM algorithm. As we have found that using much larger number of features did not improve performance, we have opted to reduce it in order to make the model more interpretable.

In addition, the author use two methods ProCTOR and QED to show that their methods are better. In the last paragraph of "Introduction", they mentioned that "To our knowledge, of the currently existing integrative methods, only ProCTOR satisfies both of these criteria, as it was specifically developed to predict failure of clinical trials. However, one of the criteria as they mention is " have no

reliance on the types of information not readily available during drug development process, like drug response human gene expression data ". I am not convinced by the way they reason that ProCTOR is the only method that they want to compare with.

We have expanded our introduction to provide more details about other methods and outlined possible issues with making comparisons in those cases. To summarize, main concern is that due to complexity of drug toxicity it can be considered from a variety of different perspectives (e.g. prediction of specific toxic responses versus prediction of clinical trial failure or development of methods applicable only to particular classes of drugs). While a method can be highly successful in the right context it will likely be suboptimal outside of it. To the best of our knowledge, this is the first study that explored the possibility to predict both market withdrawals and clinical trial failure primarily from protein binding data, and for this reason making fair and valid comparison with other methods is challenging. ProCTOR method was chosen as it is the least contextually different approach and QED as a representative high-performing method based purely on the chemical features.

Their new method uses drug target information. They argued that the issue of data sparsity and complexity can be solved by mapping drug targets onto a biological network by transforming to DSD space. I searched the whole paper, how this can be solved, theoretically, was not discussed.

We have modified our introduction to be more specific about these issues and how our approach addresses to explain this more clearly.

Drug target information was obtained from DrugBank. They should have at least mentioned/discussed that the effects from hidden off-targets.

We acknowledge that our explanation regarding the off-targets was not precise enough. Actually 'targets' as was used in the text referred to 'binding targets' (i.e. including off-targets). We fixed the ambiguity by replacing 'targets' with 'bound proteins' throughout the text and explaining that data about all drug-binding proteins (target(s) and off-targets) were used in this analysis.

I understand that they used network context as input candidate features for classifiers, but it is still confusing to read figure labels in Figure 3E and Figure 3F.

We have re-worked this figure so that the labels in question are no longer used.

In section of "model interpretation", authors discussed their predicted "toxicity risk map". From their description of the clusters of "toxicity risk map", it looks like these genes are playing very important roles in biological functions. It would be interesting to further investigate if these genes are already known to be associated with drug adverse reactions. We suggest authors to address this.

We have added an additional paragraph about specific proteins identified as particularly risk-associated by our method and how these predictions relate to current knowledge.

November 11, 2018

RE: Life Science Alliance Manuscript #LSA-2018-00098-TR

Dr. Artem Lysenko
RIKEN
Center for Integrative Medical Sciences
1-7-22 Suehiro-cho
Tsurumi
Yokohama, Kanagawa 230-0045
Japan

Dear Dr. Lysenko,

Thank you for submitting your revised manuscript entitled "An integrative machine learning approach for prediction of toxicity-related drug safety". As you will see, the reviewers appreciate the introduced changes and are now in favor of publication, pending small amendments needed to address reviewer #1's final comments.

We would thus like to invite you to submit a final version, addressing the remaining concerns. Additionally, please pay attention to the following:

- please call out the figures chronologically
- please add callouts in the manuscript text for figure panels 3A, B, C
- Figure 3 is currently listed as 'Figure 1' in the legend, please fix
- please add a legend for Tables S1, S2, S3
- please provide your ORCID iD, you should have received an email with instructions on how to do so

A. FINAL FILES:

-- High-resolution figure, supplementary figure and video files uploaded as individual files: See our detailed guidelines for preparing your production-ready images, <http://life-science-alliance.org/authorguide>

-- Summary blurb (enter in submission system): A short text summarizing in a single sentence the

study (max. 200 characters including spaces). This text is used in conjunction with the titles of papers, hence should be informative and complementary to the title. It should describe the context and significance of the findings for a general readership; it should be written in the present tense and refer to the work in the third person. Author names should not be mentioned.

B. MANUSCRIPT ORGANIZATION AND FORMATTING:

Full guidelines are available on our Instructions for Authors page, <http://life-science-alliance.org/authorguide>

Thank you for your attention to these final processing requirements.

Sincerely,

Reviewer #1 (Comments to the Authors (Required)):

Lysenko et al. have addressed the majority of our concerns raised and the manuscript is very much improved. The manuscript still requires some additional revisions:

Major Concerns:

1. The drugs used for "idiosyncratically toxic", n=38 and "HLA-toxicity" (n=9) should be included in an Appendix Table.
2. Out of the selected drugs for how many have "toxic targets" or which mechanism of toxicity have been suggested? And how many were recovered with the proposed method?
3. Figure 4D showing feature importance is not overly informative because it does not show which features are most important to evaluate "idiosyncratic toxicity". This is important to better design and evaluate future clinical studies.
4. In the discussion the authors mentioned "AKT1 is an potassium channel protein ..." which is wrong. AKT1 is a major serine-threonine kinase that regulates metabolism and signaling pathways within the cell. Please, review it carefully!
5. LYN and TLR4 were identified as highest toxicity scoring prediction targets. These results are representative of the full dataset analysis or detected only the "HLA-toxicity drugs"?

Minor concerns:

1. In Figures 1D, 5A-D "Dimention" should be substituted by "Dimension"
2. Figure legend 1 is not descriptive of figure 1. Requires additional revision.
3. Figure legends are not order properly: Figure legends 1 and 2 are followed by figure legend 1 and then figure 5.
4. Figure 5 is missing "scale" labels.

Reviewer #2 (Comments to the Authors (Required)):

The revision is acceptable.

Reviewer #3 (Comments to the Authors (Required)):

The authors have responded to our concerns very thoroughly and we are satisfied with these responses, and enthusiastic about the contribution.

Changes requested by the Reviewers

Major Concerns

1. The drugs used for "idiosyncratically toxic", n=38 and "HLA-toxicity" (n=9) should be included in an Appendix Table.

The "idiosyncratically toxic" list has been made available as Table S3 and HLA-toxicity list as Table S4.

2. Out of the selected drugs for how many have "toxic targets" or which mechanism of toxicity have been suggested? And how many were recovered with the proposed method?

In our attempts to answer this, we have searched for possible resources that collect such information and were able to find only one database (DITOP) that could have been suitable. However, DITOP has now ceased development and is no longer available. Given that our dataset has 197 toxic drugs and the complexity of the subject, curating the literature to confirm causal toxic proteins and mechanisms for all of them would involve the amount of work comparable to writing a separate review paper. Therefore unfortunately we were not able to do a comprehensive quantitative evaluation for this part of the results beyond the discussion of specific findings already included in the paper.

3. Figure 4D showing feature importance is not overly informative because it does not show which features are most important to evaluate "idiosyncratic toxicity". This is important to better design and evaluate future clinical studies.

As Figure 4 in that version did not have panels, in our answer we assumed the reviewer meant Figure 3 D. Catboost only allows importance statistic to be calculated for the complete model rather than a subset of samples. For this reason an alternative measure of importance (SHAP values) was used to address this point, which allow such granularity. We have now added an additional figure (Fig S4) showing SHAP-based feature importance summary specifically for the "idiosyncratic toxicity" subset presented in a similar way to Fig. 3 D and added our interpretation of these results to the text. A more detailed breakdown of this feature importance is also included in Table 1.

4. In the discussion the authors mentioned "AKT1 is an potassium channel protein ..." which is wrong. AKT1 is a major serine-threonine kinase that regulates metabolism and signaling pathways within the cell. Please, review it carefully!

We have changed this statement to correctly refer to AKT1 as a serine-threonine kinase

5. LYN and TLR4 were identified as highest toxicity scoring prediction targets. These results are representative of the full dataset analysis or detected only the "HLA-toxicity drugs"?

These results were for the druggable genome annotation part of the analysis. We have added an additional mention of this to the beginning of the paragraph to make this clearer.

Minor concerns

1. In Figures 1D, 5A-D "Dimention" should be substituted by "Dimension"

We have corrected all these instances

2. Figure legend 1 is not descriptive of figure 1. Requires additional revision.

We have split up the figure into several parts and revised all of the legends to be more appropriate.

3. Figure legends are not order properly: Figure legends 1 and 2 are followed by figure legend 1 and then figure 5.

Figure title was corrected.

4. Figure 5 is missing "scale" labels.

Label for the scale bar has now been added

November 20, 2018

RE: Life Science Alliance Manuscript #LSA-2018-00098-TRR

Dr. Artem Lysenko
RIKEN
Center for Integrative Medical Sciences
1-7-22 Suehiro-cho
Tsurumi
Yokohama, Kanagawa 230-0045
Japan

Dear Dr. Lysenko,

Thank you for submitting your Methods entitled "An integrative machine learning approach for prediction of toxicity-related drug safety". I appreciate the introduced changes, and it is a pleasure to let you know that your manuscript is now accepted for publication in Life Science Alliance. Congratulations on this interesting work.

DISTRIBUTION OF MATERIALS:

Again, congratulations on a very nice paper. I hope you found the review process to be constructive and are pleased with how the manuscript was handled editorially. We look forward to future exciting submissions from your lab.

Sincerely,

Andrea Leibfried, PhD
Executive Editor

Life Science Alliance
Meyerhofstr. 1
69117 Heidelberg, Germany
t +49 6221 8891 502
e a.leibfried@life-science-alliance.org
www.life-science-alliance.org